# NR2F1, a Tumor Dormancy Marker, Is Expressed Predominantly in Cancer-Associated Fibroblasts and Is Associated with Suppressed Breast Cancer Cell Proliferation

**DOI:** 10.3390/cancers14122962

**Published:** 2022-06-15

**Authors:** Rongrong Wu, Arya Mariam Roy, Yoshihisa Tokumaru, Shipra Gandhi, Mariko Asaoka, Masanori Oshi, Li Yan, Takashi Ishikawa, Kazuaki Takabe

**Affiliations:** 1Department of Surgical Oncology, Roswell Park Comprehensive Cancer Center, Buffalo, NY 14263, USA; rongrong.wu@roswellpark.org; 2Department of Breast Surgery and Oncology, Tokyo Medical University, Tokyo 160-8402, Japan; Masaoka@tokyo-med.ac.jp (M.A.); tishik@tokyo-med.ac.jp (T.I.); 3Department of Medicine, Roswell Park Comprehensive Cancer Center, Buffalo, NY 14263, USA; arya.roy@roswellpark.org (A.M.R.); shipra.gandhi@roswellpark.org (S.G.); 4Department of Surgical Oncology, Graduate School of Medicine, Gifu University, Gifu 501-1193, Japan; yt1090@gifu-u.ac.jp; 5Department of Gastroenterological Surgery, Graduate School of Medicine, Yokohama City University, Yokohama 236-0004, Japan; oshi.mas.wc@yokohama-cu.ac.jp; 6Department of Biostatistics & Bioinformatics, Roswell Park Comprehensive Cancer Center, Buffalo, NY 14263, USA; li.yan@roswellpark.org; 7Department of Surgery, Jacobs School of Medicine and Biomedical Sciences, University at Buffalo, Buffalo, NY 14263, USA; 8Department of Surgery, Graduate School of Medical and Dental Sciences, Niigata University, Niigata 951-8510, Japan; 9Department of Breast Surgery, Fukushima Medical University, Fukushima 960-1295, Japan

**Keywords:** breast cancer, cancer associated fibroblasts, dormancy, NR2F1, single cell sequence, TGF beta

## Abstract

**Simple Summary:**

Tumor dormancy is a state in which some cancer cells are dormant by less cell proliferation, and are resistant to treatments, thus leading to late recurrence. As nuclear receptor subfamily 2 group F member 1 (NR2F1) is a biomarker of dormancy shown in experimental settings, we studied its clinical relevance by analyzing comprehensive gene expression profiles of approximately 7000 breast cancer patients using computational biological approaches. *NR2F1* gene expression of bulk tumors correlated with the suppression of cell proliferation and metastasis-related pathways, but not with metastasis or survival. Using single-cell sequence cohorts, we found that *NR2F1* was predominantly expressed in cancer-associated fibroblasts, particularly in inflammatory type, rather than in cancer cells. Further, *NR2F1* expression in breast cancer cells did not consistently correlate with stem cell-like traits. Our findings suggest that the *NR2F1* gene expression signal from a patient’s bulk tumor is not from the cancer cells.

**Abstract:**

Background: Tumor dormancy is a crucial mechanism responsible for the late recurrence of breast cancer. Thus, we investigated the clinical relevance of the expression of *NR2F1*, a known dormancy biomarker. Methods: A total of 6758 transcriptomes of bulk tumors from multiple breast cancer patient cohorts and two single-cell sequence cohorts were analyzed. Results: Breast cancer (BC) with high *NR2F1* expression enriched TGFβ signaling, multiple metastases, and stem cell-related pathways. Cell proliferation-related gene sets were suppressed, and MKi67 expression was lower in high *NR2F1* BC. In tumors with high Nottingham grade, *NR2F1* expression was found to be lower. There was no consistent relationship between *NR2F1* expression and metastasis or survival. Cancer mutation rates, immune responses, and immune cell infiltrations were lower in high *NR2F1* tumors, whereas the infiltration of stromal cells including cancer-associated fibroblasts (CAFs) was higher. *NR2F1* was predominantly expressed in CAFs, particularly inflammatory CAFs, rather than in cancer cells, consistently in the two single-cell sequence cohorts. Conclusions: *NR2F1* expression in breast cancer is associated with tumor dormancy traits, and it is predominantly expressed in CAFs in the tumor microenvironment.

## 1. Introduction

Breast cancer is the most common cancer in women, and the vast majority of the deaths occur after cancer cells metastasize systemically to bone, liver, lung, and brain long after the completion of initial treatment, also known as late recurrence [1]. In particular, estrogen receptor (ER)-positive breast cancers are known to develop late recurrence mainly in the bone with about half of the recurrences occurring 5 years after the initial diagnosis of the primary tumor [2,3]. Adjuvant chemotherapy and 5 years of adjuvant endocrine therapy have greatly reduced the overall recurrence rates [1]; however, late recurrence has emerged as an issue given the longevity of the population. Low recurrence rates of about 0.5% to 3% are observed for the first few years after the adjuvant treatments [4,5], and eventually, more than half of breast cancer patients die from late recurrence after 5 to 20 years of initial treatment [6]. This prolonged latency occurs due to cancer cells surviving in a non-proliferative cell cycle arrest, called tumor dormancy [2,6]. These dormant cancer cells remain in a quiescent state, thus evading cytotoxic chemotherapy and apoptosis that target proliferating cells [7,8,9] and hide undetected for decades and regain proliferation that surfaces as late recurrence at some point with unknown triggers [10]. To this end, tumor dormancy is considered one of the major mechanisms of drug resistance that leads to patient deaths [11].

It is well known that even early-stage breast cancer can scatter disseminated tumor cells (DTCs) [12], which can invade, spread, resist chemotherapy, and switch to quiescent states [10]. These characteristics resemble tumor dormancy cells where some were shown to express TGFβ that regulates cancer stem cells [13,14,15]. The activation of cellular plasticity pathways such as Wnt or Rank was shown to be necessary for dormant cell dissemination in primary breast cancer [16]. Furthermore, dormant DTCs recapitulate the quiescent program of normal stem cells, regulating pluripotency and reducing proliferation through BMP and TGFβ family members for a long period [17].

Nuclear receptor subfamily 2 group F member 1 (NR2F1) is an established biomarker of tumor dormancy that promotes quiescence of various types of cancer cells, including breast, prostate, and squamous cell carcinoma of the head and neck [18,19]. NR2F1 is upregulated in bone marrow DTCs via SOX9 and RARβ and regulates the expression of genes related to quiescence, cell survival, and pluripotency [18]. High NR2F1 expression in the bone marrow of DTC-positive breast cancer was associated with a better prognosis, indicating that NR2F1 induces durable growth arrest in metastatic cells [20]. Administration of NR2F1 agonist to the primary squamous cell carcinoma of the head and neck induced tumor dormancy and suppressed metastasis [21].

The dormancy of disseminated cells at latent sites is considered undesirable, as it leads to late recurrence upon reactivation; however, high expression of *NR2F1* that prolongs dormancy may keep those cells from reactivation and thus recurrence. We hypothesize that high expression of the *NR2F1* gene in the primary breast cancer tumor microenvironment (TME) maintains the quiescence of cancer cells and is associated with clinical outcomes in breast cancer patients. We have investigated cell fractionation and immune activation in the TME using large public patient cohorts in silico using the techniques we have previously described [22,23,24,25]. To the best of our knowledge, this is the first translational study to investigate the clinical relevance of *NR2F1* gene expression in primary breast cancer.

## 2. Materials and Methods

### 2.1. Collection of Breast Cancer Sample Data

The transcriptome of the bulk primary breast tumor of 6245 patients and 512 metastatic breast cancer patients and single-cell sequence data of 20 primary breast cancer patients were analyzed. For large primary breast cancer cohorts, curated RNA-seq data of the bulk tumors from the Cancer Genome Atlas (TCGA, *n* = 1077) [26] and Molecular Taxonomy of Breast Cancer International Consortium (METABRIC, *n* = 1904) [27] were downloaded from cBiopotal [28], and GSE96058 (*n* = 3069) [29,30] from the Gene Expression Omnibus (GEO) database, as previously reported [24,31,32]. Nottingham histological grade of TCGA was manually calculated from pathology reports of TCGA-BRCA as previously reported [33]. We also downloaded the following curated RNA-seq data of the bulk tumors from GEO database: GSE12276 (*n* = 195) [34] of early breast cancer with known later metastatic sites, GSE124647 (*n* = 140) [35] and GSE159956 (*n* = 295) [36] of advanced breast cancer, GSE110590 (*n* = 77) [37] of primary and metastatic samples of advanced breast cancer, and GSE180280 (*n* = 37) [38], GSE87455 (*n* = 153) [39,40,41], GSE28844 (*n* = 61) [42], GSE21974 (*n* = 57) [43] as cohorts including before and after neoadjuvant chemotherapy samples of primary breast cancer. Matching probes were used for *NR2F1* expression, and the average of the probes was used when there were multiple probes. The R package GEOquary was used to acquire and process data from GEO. We also used RNA-seq data of GSE172882 [44], a cohort of dormant D2.OR and proliferate D2A1 breast cancer cell lines, as well as 4T1 and metastasis-competent derivative cell line 4T1.2 from our previous publication [45].

Two cohorts of single-cell seq data were downloaded from the Single-cell Portal (https://singlecell.broadinstitute.org/single_cell, accessed on 21 November 2021): 5 triple-negative breast cancer (TNBC) samples reported by Wu et al. in 2020 [46] (Cohort 1) and 26 primary breast cancer tumors including 11 estrogen receptor (ER) positive, 5 human epidermal growth factor receptor (HER) 2 positive and 10 TNBCs reported by Wu et al. in 2021 [47] (Cohort 2).

### 2.2. Gene Set Enrichment Analysis

For functional analysis of the bulk RNA sequencing data, Gene set enrichment analysis (GSEA) [48,49] was performed comparing “high” and “low” groups divided by the median *NR2F1* expression. The free software GSEA v4.2.3 for Windows was used to analyze 50 gene sets of Hallmark from the Molecular Signatures Database [50]. As previously reported [51,52], the normalized enrichment score (NES) was utilized for assessment and the false discovery rate (FDR) of less than 0.25 was considered significant. For single-cell seq data, GSEA was conducted exclusively on cancer cells. Since most cells had no expression of *NR2F1*, we compared cells with positive and negative *NR2F1* expression in cancer cells. Seurat and fgsea R package for the R platform was used for analyses.

### 2.3. Intertumoral Cell Fraction Estimation

Intratumor microenvironment deconvolution of the bulk tumor was performed using xCell [53] to analyze immune and stromal cell fractions, as previously reported [23,51,54]. The geometric mean of Perforin 1 and granzyme A expression level was used to calculate cytolytic activity (CYT) to quantify cell-killing immunity [55,56]. Several score values of TCGA patients reported by Thorsson et al. in 2018 [57] were used: proliferation score, intra-tumor heterogeneity, homologous recombination loss, silent and non-silent mutation rates, interferon response, fraction altered, tumor-infiltrating lymphocyte fraction, stromal fraction.

### 2.4. Others

The analysis and visualization of this study were performed using R (version 4.0.1). The “high” and “low” *NR2F1* groups of TCGA, METABRIC, and GSE96058 were defined by median expression of *NR2F1* as cutoff as explained above. All analyses of single-cell seq data were performed on the R platform using the VlnPlot and DotPlot commands from the R package Seurat. The classification of cell types is different in Cohort 1 and in Cohort 2, and the cell annotations for single-cell seq data are obtained from the original paper published by the authors. Overall survival (OS) was defined as the time from the end of therapy to death from any cause, disease-specific survival (DSS) was defined as the time to death from breast cancer, and disease-free survival (DFS) was defined as the time to recurrence.

## 3. Results

### 3.1. Tumor Dormancy-Related Genes (NR2F1, TGFβ, RARβ) Were Highly Expressed in the Dormant and Metastatic Cell Lines, and High NR2F1 Breast Cancer Enriched Multiple Cancer Metastasis-Related Pathways including TGFβ Signaling Gene Set

NR2F1 is a known tumor dormancy marker, and RARβ and TGF-β signaling have been reported to induce dormancy [13,15]. We demonstrated that the expression of *NR2F1* gene was higher in previously established dormant cells (D2OR murine breast cancer cells) [44] compared to the proliferative cells (D2A1 cells) in 3D cultures, and the expression of *RARB* and *TGFB1* genes are higher in dormant cells compared to the proliferative cells in both 2D and 3D cultures (Figure 1A, all *p* < 0.02). Further, both *NR2F1* and *TGFB1* gene expressions were significantly higher in bone metastatic derivatives [45] compared with wild-type 4T1 murine breast cancer cells (Figure 1B). TGF-β signaling calculated by Thorsson et al. [57] in the TCGA cohort was higher in the *NR2F1* high group (Figure 1C). This result was validated by gene enrichment analysis (GSEA) that showed enrichment of TGF-β signaling in the *NR2F1* high group consistently in the three independent large primary breast cancer cohorts (TCGA, METABRIC, and GSE96058; Figure 1D). *NR2F1*-high primary breast cancer amplified multiple pathways related to metastasis: Epithelial Mesenchymal Transition (EMT), angiogenesis, as well as enhanced cancer stem cell-related pathways, including KRAS signaling, Notch signaling, Hedgehog signaling, and Wnt/β-catenin signaling (Figure 1D). These results confirm that *NR2F1* gene expression is associated with dormancy, metastasis, TGF-β signaling, as well as with the cancer stem cell phenotype that has similar biological characteristics to tumor dormancy.

### 3.2. Cell Proliferation Was Consistently Suppressed in Breast Cancer with High NR2F1 Expression

Given that cell growth arrest is the major characteristic of tumor dormancy, it was of interest to investigate the relationship between *NR2F1* expression and cell proliferation. The association of *NR2F1* levels with the Nottingham histological grades and the cell proliferation marker *MKI67* was studied for the pathological assessment of cancer cell proliferation. We found that Nottingham grade was inversely related to *NR2F1* expression in TCGA and METABRIC, but not in GSE96058 (Figure 2A). *MKI67* expression was lower in the *NR2F1* high tumor consistently in all three cohorts, and the proliferation score calculated by Thorsson et al. was also lower in TCGA (Figure 2A). Five cell proliferation-related gene sets defined in the Hallmark collection (E2F targets, G2M checkpoint, Myc targets v1 and v2, and Mitotic spindle,) were all strongly suppressed in the high *NR2F1* group in all cohorts (Figure 2B). These results suggest that *NR2F1* expression in primary breast cancer is associated with low cell proliferation.

### 3.3. NR2F1 Expression in Primary Breast Cancer Was Higher with Lymph Node Metastasis but Was Not Consistently Associated with Distant Metastasis or Survival

When *NR2F1* expression is associated with metastatic disease, it is of interest to study its relationship with clinical outcomes. Therefore, we investigated the association of *NR2F1* expression with breast cancer survival. The survival analysis comparing the high and low *NR2F1* expression in primary breast cancer did not show consistent results. The high-*NR2F1* group showed worse disease-specific (DSS) and disease-free survival (DFS) in the METABRIC cohort, while it showed better overall survival (OS) in GSE96058. None of these results were validated by another cohort (Figure 3A). Given that breast cancer is a heterogenous disease, it was of interest to investigate the survival outcome by *NR2F1* expression of each subtype. The OS of *NR2F1* high triple-negative and ER-positive/HER2-negative subtypes was associated with better survival in GSE96058, and DSS of *NR2F1* high HER2-positive subtypes was associated with worse survival in METABRIC. However, none of these results were validated by the other breast cancer cohorts (Appendix A). We investigated the association of *NR2F1* expression with metastasis. No significant difference was found in the *NR2F1* expression with and without distant metastasis in five independent primary breast cancer cohorts (Figure 3B). Conversely, *NR2F1* expression was higher in the group with lymph node metastasis in three out of four cohorts (Figure 3C). In addition, there was no difference in NR2F1 expression in primary breast cancer based on the known site of relapse (Figure 3D). It was also of interest to compare the expression of *NR2F1* between primary and metastatic breast cancer, but there was no difference observed in the *NR2F1* expression between both groups (Figure 3E). Based on the above results, we concluded that there were no differences in the survival or metastasis by *NR2F1* expression levels in the patient cohorts.

### 3.4. Cancer Cells That Express NR2F1 Were Not Associated with Upregulation of Metastasis-Related Gene Sets or with Downregulation of Cell Proliferation-Related Gene Sets

The transcriptome of the patient cohorts was obtained from bulk tumors, which is a mixture of messenger-RNA expression signals from all cell types in the TME. Therefore, it was of interest to investigate the pathway that is involved with *NR2F1* expression solely within cancer cells. We used two primary breast cancer single-cell sequencing cohorts, Cohort 1 and 2 [46,47], where 5.1% and 2.7% of cancer cells excluding the other cell types expressed any *NR2F1* mRNA (*NR2F1*-positive group), respectively. GSEA showed minimal concordance between the two cohorts in the *NR2F1*-positive group (Figure 4). In contrast to the bulk tumor results, TGFβ and cancer stem cell-related pathways were not enriched in *NR2F1*-positive cancer cells, and conversely, TGFβ signaling and Wnt/β-catenin signaling gene sets were suppressed in Cohort 1 (Figure 4 left). Cell proliferation pathways such as E2F targets, Myc targets v1, G2M checkpoints, and mTOR signaling were suppressed in *NR2F1*-positive cancer cells of Cohort 1 but were enriched in that of Cohort 2. EMT gene set was enriched in the *NR2F1*-positive group in Cohort 2, but not in Cohort 1. These results show that cancer cells from breast cancer patients that express *NR2F1* were not consistently associated with common pathways seen in tumor dormancy.

### 3.5. High Expression of NR2F1 Was Associated with a Lower Mutational Rate, Less Immune Response, and Less Infiltration of Immune Cells, but a Higher Abundance of Stromal Cells

Since breast cancer cells in patients that express any *NR2F1* did not consistently demonstrate the phenotype of dormancy, we investigated the association of *NR2F1* expression and cancer cell mutation rates, as well as the cell fractions that constitute the TME of the *NR2F1* high expression breast cancer. Intratumoral heterogeneity, homologous recombination deficiency (HRD), silent and non-silent mutation rates were all decreased in *NR2F1*-high tumors in TCGA (Figure 5A). Cancer immune activity measured by interferon (IFN)-γ response, fractional change, and tumor-infiltrating lymphocytes (TIL) fraction were all significantly lower in the *NR2F1*-high tumors in TCGA (Figure 5A). In agreement, the fraction of immune cells such as T helper type 1 (Th1) cells, T helper type 2 (Th2) cells, and M1 and M2 macrophages was significantly lower in the *NR2F1*-high tumors consistently in GSE96058, METABRIC, and TCGA cohorts (Figure 5B). Conversely, stromal cells such as adipocytes, fibroblasts, mesangial cells, and endothelial cells were all significantly abundant in *NR2F1*-high tumors consistently in TCGA, METABRIC, and GSE96058 cohorts (Figure 5B). Further analysis by each immunohistological subtype showed the same trend that the fraction of stromal cells was significantly higher in the *NR2F1* high group across multiple cohorts (Appendix A). These data suggest that *NR2F1*-high breast cancers have less immune response, less immune cell infiltration, and a higher density of stromal cells.

### 3.6. Cancer-Associated Fibroblasts (CAF), Not Cancer Cells, Were the Major Contributor to the NR2F1 Expression Signal in a Bulk Breast Tumor

Given the high infiltration of stromal cells in the *NR2F1*-high tumors, we investigated the cell type that contributes to the *NR2F1* gene expression signal using single-cell sequencing cohorts. In Cohort 1, cancer-associated fibroblasts (CAFs) were the predominant cells followed by cancer (epithelial) cells that expressed *NR2F1* among all the cell types, as shown in Figure 6A (left). Unlike for *NR2F1*, which was almost exclusively expressed in CAFs, for the other dormancy-related genes: *TGFB1* was predominantly expressed by myeloid cells followed by endothelial cells and lymphocytes, *SOX9* was predominantly expressed in cancer (epithelial) cells followed by normal (myo)epithelial cells, and *RARB* was expressed weakly in normal (myo)epithelial cells, CAFs and perivascular-like cells (PVL) (Figure 6A right). These results were strikingly mirrored in Cohort 2 (Figure 6B), and they were similar regardless of the subtypes in Cohort 2 (Appendix A).

### 3.7. NR2F1 Was More Highly Expressed in Inflammatory CAFs, and in CAFs with Lymph Node Metastasis and in More Advanced Stages, but No Differences Were Observed in Cancer Cells

Since CAFs were the major contributor to *NR2F1* gene expression, we further investigated the types of CAFs that express *NR2F1*. *NR2F1* was found to be consistently highly expressed in inflammatory CAFs, compared to myofibroblastic CAFs in both Cohort 1 and Cohort 2 (Figure 7A), regardless of the subtypes (Appendix A). 

Next, it was of interest to investigate the expression levels of tumor dormancy markers based on the cancer progression (tumor size and lymph node metastasis) and based on the effect of chemotherapy on cancer (epithelial) cells and CAFs. Single-cell sequence Cohort 2 was analyzed, as it contains clinical information. Levels of expression of tumor dormancy markers in primary breast cancer were compared before and after neoadjuvant chemotherapy by anthracyclines and taxanes in four cohorts. Few cancer (epithelial) cells expressed *NR2F1* and *TGFB1*. *SOX9* was highly expressed in the T4 or tumors with lymph node metastasis or without treatment (Figure 7B upper). Conversely, in CAFs, *NR2F1* expression was higher in T3 or tumors with lymph node metastasis, *TGFB1* was higher in larger T categories or without lymph node metastasis, and there was little difference in *SOX9* expression (Figure 7B). *NR2F1* expression was higher in treatment naïve than in treated CAFs, which was opposite the results that were observed in cancer cells (Figure 7B). Little *RARB* expression was detected in either type of cell. See Appendix A for violin plots of *NR2F1*, *TGFB1*, *SOX9*, and *RARB* expression in cancer cells and CAFs. In summary, we found that higher *NR2F1* expression in CAFs was associated with tumors of larger size and with lymph node metastasis. *NR2F1* expression was significantly higher after treatment in two out of four neoadjuvant cohorts (Figure 7C). There was no difference in *NR2F1* expression between responder and non-responder to neoadjuvant endocrine therapy (Appendix A). The details of neoadjuvant treatment for each cohort are summarized in Appendix A.

## 4. Discussion

In this study, we demonstrated that *NR2F1* gene expression in a bulk breast tumor is associated with metastasis-related as well as stem cell-related pathways, including suppression of cell proliferation, but it is not associated with distant metastasis or survival. We also found that *NR2F1* is most strongly expressed in cancer-associated fibroblasts (CAFs) and not in cancer cells in the tumor microenvironment (TME). NR2F1 is a known marker of tumor dormancy and is expressed higher in dormant cell lines and bone metastatic clone cells. The cancer stem cell-related pathways, as well as metastasis-related pathways, were enriched in the *NR2F1*-high expressing bulk breast tumor. *NR2F1* expression was lower in tumors with high Nottingham grade, high Ki67 expression, and high proliferation scores. All five cell proliferation-related pathways from the Hallmark collection were less enriched in the high *NR2F1* tumors. All these results indicated that *NR2F1* expression was associated with suppressed cell proliferation, which agrees with the notion that NR2F1 is a tumor dormancy marker. Conversely, there was no consistent association between *NR2F1* expression and survival among the three cohorts, and no coherent relationship was observed between *NR2F1* expression in primary breast cancer and distant metastases. Therefore, we performed gene set enrichment analysis exclusively on cancer cells from single-cell sequence cohorts and found that there was no consistent enrichment of metastasis-related, stem cell-related gene sets or suppression of cell proliferation-related gene sets in *NR2F1* high cancer cells. There was no difference in mutation rates by the level of *NR2F1* expression in a bulk tumor, however; immune-related scores such as IFN-γ response and TIL fractionation, as well as immune cell infiltrations such as Th1 and Th2 T cells and M1 and M2 macrophages, were all less in *NR2F1* high expression breast tumors. On the contrary, stromal cells such as adipocytes, fibroblasts, and endothelial cells were more abundant in these tumors. *NR2F1* was rarely expressed in cancer cells and was most prominently expressed in CAFs, as assessed by cell types in a single-cell sequence cohort and validated in another cohort. Among the CAF types, *NR2F1* expression was higher in inflammatory CAF than in myofibroblastic CAF as well as in larger-size tumors and tumors with lymph node metastasis. Finally, *NR2F1* expression was higher in CAFs without treatment compared to those treated with chemotherapy, whereas *NR2F1* expression was higher in cancer cells that underwent treatment compared to those that are not treated. There was no consistent increase in *NR2F1* expression after treatment in bulk tumors.

NR2F1 is an orphan nuclear receptor [58] that regulates enhancer elements during human neural crest cell differentiation [59], which also interacts with chromatin remodeling enzymes and mediates histone modifications [60]. Because of this pleiotropic regulation and proliferation suppression, NR2F1 has been identified as a master regulator of tumor dormancy [21]. SOX9, RARβ, and TGF-β are key genes that are related to dormancy as well as to Wnt and Notch pathways [18,61]. The *NR2F1* high-expression bulk breast tumors in the cohorts we analyzed were consistent with suppressed cell proliferation, higher expression of dormancy-related genes, and enrichment of pathways as observed in the previous reports. Dormant cells can originate and disseminate even from early-stage cancers [10,62]. In agreement, we observed that the EMT pathway is enriched in the high *NR2F1* tumors. In addition, the *NR2F1* high group showed lower mutation burden, HRD with decreased proliferation, and less immune response and immune cell infiltration, indicating immune “cold” tumors. Conversely, residual dormant tumor cells after treatment were reported to be associated with tumor heterogeneity [63], whereas intratumor heterogeneity was less in the high *NR2F1*-expressing breast tumors in TCGA.

Despite high *NR2F1* tumors showing tumor dormancy traits consistently, *NR2F1* expression was not related to survival or distant metastasis. Single-cell sequencing showed that *NR2F1* was almost exclusively expressed in CAFs, and rarely in cancer cells. The transcriptome of a bulk tumor is an average of the mRNA expression in the TME. Cell types other than cancer cells in the TME can be responsible for a gene expression signature. This is in agreement with our results showing that cancer cells alone cannot enrich stem cells and metastasis-related pathways. Unlike mesenchymal markers such as αSMA, fibroblast-specific protein 1 (FSP1), and fibroblast activation protein (FAP) [64], NR2F1 has not been described to be highly expressed in breast cancer CAFs. Through interactions with several types of cells in the TME, mesenchymal stromal cells including fibroblasts may either promote or inhibit tumor growth [65,66]. This is because CAFs are a heterogeneous population of cells that can be divided into subsets with different functions. In the single-cell cohort used in this study, CAFs were classified into inflammatory CAFs (iCAFs) and myofibroblast-like CAFs (myCAFs), and *NR2F1* was significantly expressed in iCAFs in both cohorts. As high *NR2F1* expression is more predominantly seen in iCAFs, which are considered more “stem cell-like” since they secrete a variety of inflammatory cytokines compared to myCAFs, this can be speculated as a reason for the enrichment of multiple stem cell-related pathways in high *NR2F1* tumors [46].

The current study demonstrated that *NR2F1* expression in the bulk tumor of primary breast cancer is associated with decreased cell proliferation and cancer stem cell-like characteristics. Previous studies mentioned that *NR2F1* was highly expressed in DTCs [67] We found that *NR2F1* is most predominantly expressed in CAFs in the TME of primary breast cancer. However, the findings from our study are not ample enough to substantiate that CAF-expressed *NR2F1* regulates breast tumor dormancy. The Aguirre-Ghiso group recently demonstrated that *NR2F1* agonist treatment induced cancer cell dormancy [21], which raised an expectation that *NR2F1* expression in primary breast cancer may have the possibility to be a biomarker. The novelty of our study is that it is CAFs, not cancer cells, that are the dominant source of *NR2F1* expression in the bulk tumor. To this end, we believe that *NR2F1* expression in the bulk tumor does not reflect expression in cancer cells; thus, its value as a biomarker is in doubt. *NR2F1* expression in cancer cells of primary breast cancer was not associated with cancer stem cell-like characteristics at all. In order to prove that CAF-expressed *NR2F1* regulates breast tumor dormancy, one needs to analyze the single-cell sequencing data and show whether the late recurrence in other organs, such as lungs, bones, and brain, other than lymph nodes, is correlated with *NR2F1* expression in CAFs.

Our study has some limitations. A major limitation is that it is a retrospective study with potential selection bias, particularly regarding clinical factors, which include data from decades ago and therefore fails to include the prognostic impact of recent treatments. Furthermore, variations among bulk tumor samples of the cohorts may result in sampling bias due to differences in the proportion of the stromal area. For single-cell cohorts, we used cell annotations at the time of cohort publication; therefore, it includes the resolution limitations of single-cell annotation techniques at that time. Finally, all the data are derived in silico analysis, and as the study does not include experimental validation of the data, we cannot prove any cause–effect relationships in TME.

## 5. Conclusions

The tumor dormancy marker *NR2F1* expression in primary breast cancer is associated with less cell proliferation, and it is predominantly expressed in cancer-associated fibroblasts in the tumor microenvironment.

## Figures and Tables

**Figure 1 cancers-14-02962-f001:**
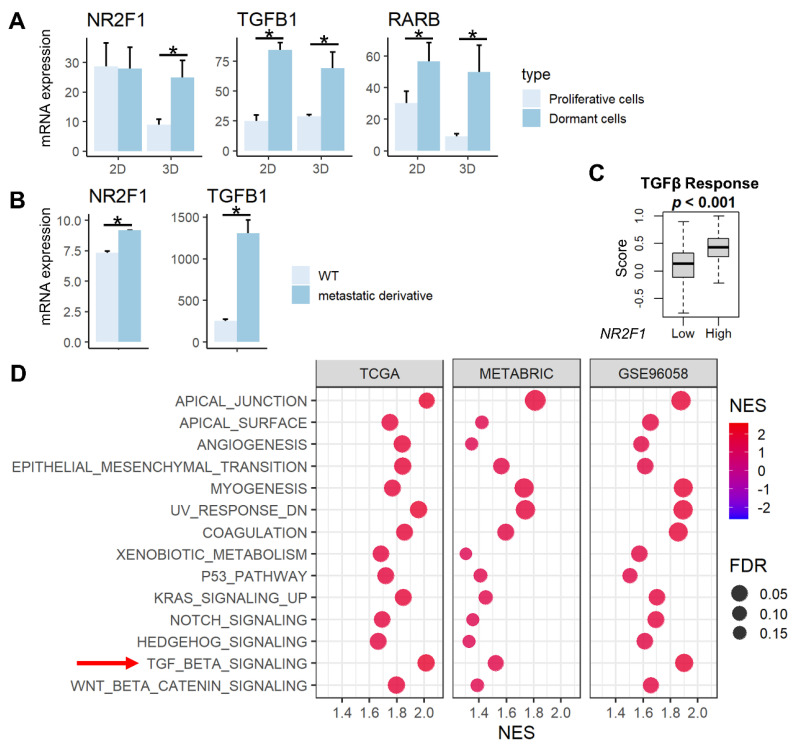
Association between *NR2F1* and TGFβ and pathways enriched in high *NR2F1* tumors. (**A**) Barplots showing mRNA expression of *NR2F1*, *TGFB1*, and *RARB* in the dormant D2OR cells (dark blue) and proliferative D2A1 cells (light blue) cultured in 2D and 3D. (**B**) Barplots showing mRNA expression of *NR2F1* and *TGFB1* in 4T1 murine breast cancer cell line and metastatic derivative 4T1.2 cell line. ANOVA test was performed between the two groups (**A**,**B**), with * indicating *p* < 0.05. (**C**) The box plot shows the TGFβ score for high and low *NR2F1* groups in TCGA. Cohorts were divided into high and low groups by median expression of *NR2F1*. Mann−Whitney U test was used to compare the two groups. (**D**) Dot plots showing hallmark gene sets that are significantly enriched in high *NR2F1* groups by gene set enrichment analysis in the TCGA, METABRIC, and GSE96058 cohorts. TGFβ gene set is highlighted by the red arrow. The size of each dot represents FDR, and the *x*−axis and color tone represent NES.

**Figure 2 cancers-14-02962-f002:**
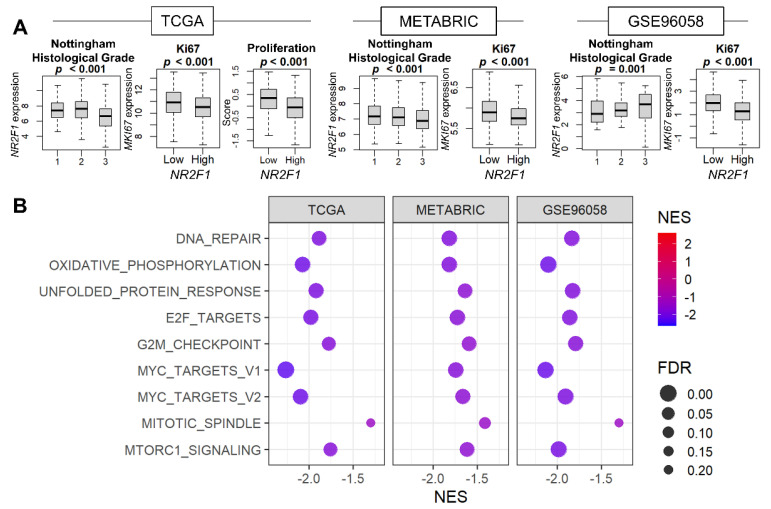
Association of *NR2F1* expression with cell proliferation−related measurements and pathways. (**A**) Boxplots of *NR2F1* expression by Nottingham histological grade and MKi67 expression in three cohorts. Proliferation score based on *NR2F1* high vs. low expression in TCGA cohort. The Kruskal–Wallis H test was used for three groups, and the Mann–Whitney U test was used for two−group comparison. (**B**) Hallmark gene sets were significantly enriched in the low *NR2F1* group in all three cohorts by GSEA.

**Figure 3 cancers-14-02962-f003:**
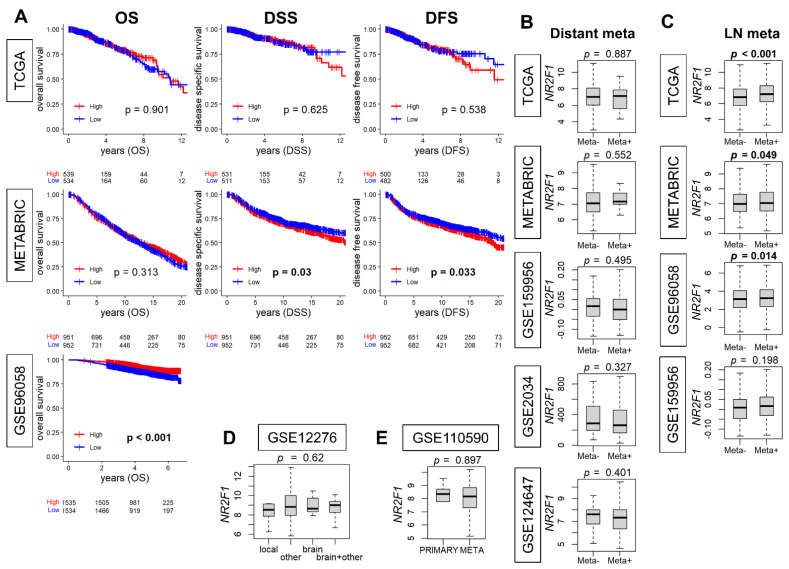
Association of *NR2F1* with clinical parameters. (**A**) Kaplan–Meier curves of overall survival (OS), disease−specific survival (DSS), and disease−free survival (DFS) based on the high and low *NR2F1* expression in three large cohorts. Log−rank test was used for the analysis, and significant p values are shown in bold. (**B**) Boxplots showing *NR2F1* expression between patients with (Meta+) and without (Meta−) distant metastasis in the five cohorts. (**C**) Boxplots showing *NR2F1* expression between patients with (Meta+) and without (Meta−) lymph node metastasis in the four cohorts. (**D**) The boxplot shows *NR2F1* expression of primary breast cancer by the known site of relapse. (**E**) The boxplot shows *NR2F1* expression of primary and metastatic breast cancer. Cohort names are indicated in the figure, and Mann–Whitney U and Kruskal–Wallis H tests were used for analysis. Significant *p* values are shown in bold. *p* < 0.05 was considered significant.

**Figure 4 cancers-14-02962-f004:**
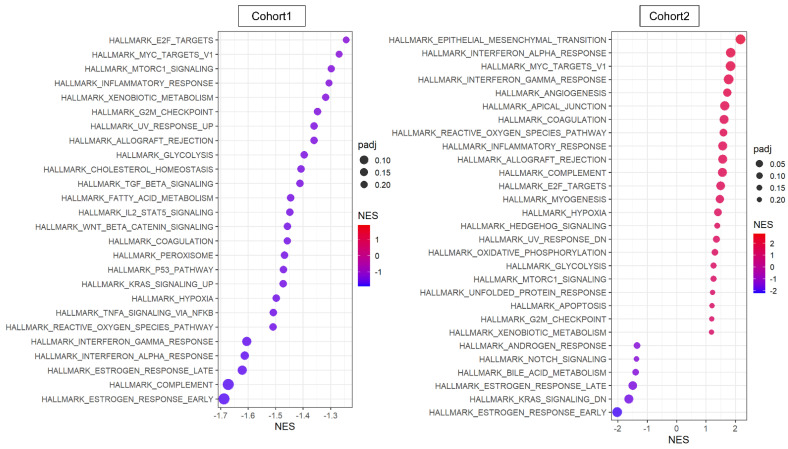
Pathway analysis of cancer cells that expressed any amount of *NR2F1*. Dot plots showing GSEA comparing cancer cells with positive and negative *NR2F1* expression in single−cell Cohort 1 (**left**) and Cohort 2 (**right**). Only significant gene sets from HALLMARK are shown in the figure. The size of each dot represents adjusted *p* value, and the x−axis and color tone represent NES.

**Figure 5 cancers-14-02962-f005:**
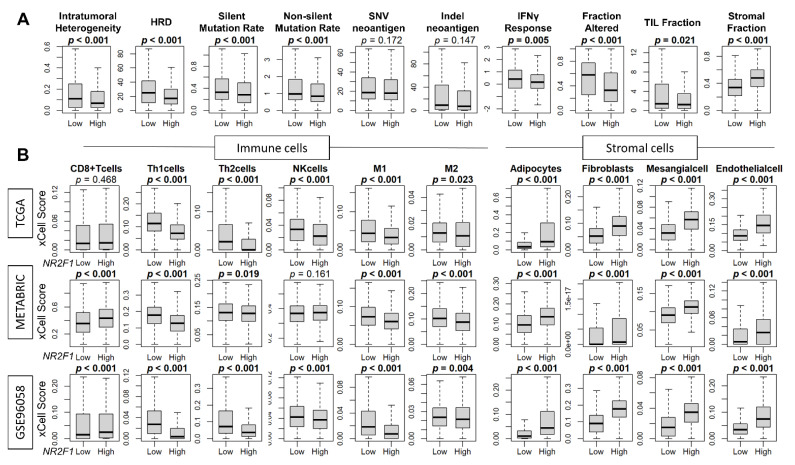
Association of *NR2F1* with immunity within the tumor microenvironment. (**A**) Boxplots showing various scores based on high and low *NR2F1* expression in TCGA. Intratumoral heterogeneity, homologous recombination deficiency (HRD), silent/non-silent mutation rate, SNV/Indel neoantigen, interferon gamma response, fraction altered, tumor-infiltrating lymphocytes (TIL) fraction, and stromal fraction. (**B**) Boxplots showing immune and stromal cell fractions between *NR2F1* high and low groups in three large cohorts. Mann–Whitney U test was used to compare the two groups, and *p* values are shown in bold for significant results (*p* < 0.05).

**Figure 6 cancers-14-02962-f006:**
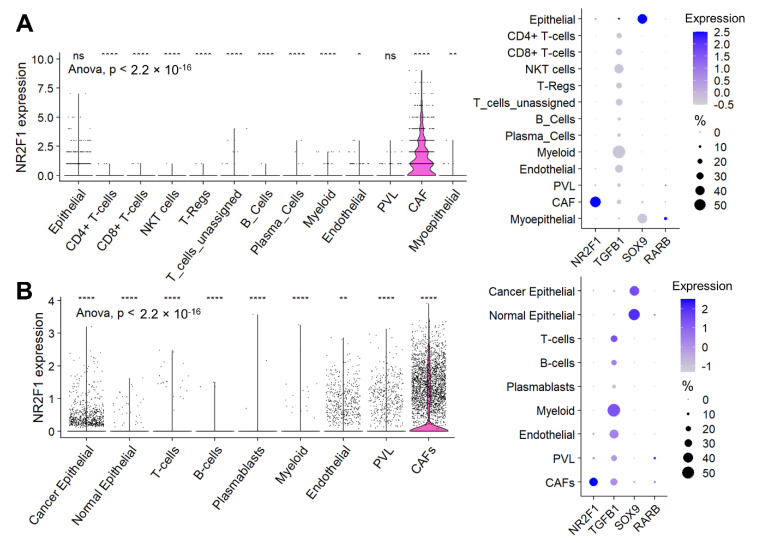
Expression of *NR2F1* and dormancy−related genes by cell types. (**A**) The violin plot on the left shows *NR2F1* expression of single-cell Cohort 1 by cell type: epithelial cells, CD4+ T-cells, CD8+ T-cells, natural killer T-cells (NKT), regulatory T-cells (T-reg), unassigned T-cells, B-cells, plasma cells, myeloid cells, endothelial cells, perivascular like cells (PVL), cancer-associated fibroblast (CAFs), and myoepithelial cells. One dot represents one cell. After multi-group comparison, baseline *NR2F1* expression and expression in each cell type were compared in two groups by one-way ANOVA test. The dot plot on the right shows *NR2F1*, *TGFB1*, *SOX9*, and *RARB* expression by cell type in the single-cell Cohort 1. The size of each dot indicates the number of cells, and the purple intensity indicates the expression level. (**B**) The violin plot on the left shows *NR2F1* expression of single-cell Cohort 2 by cell type: cancer epithelial cells, normal epithelial cells, T cells, B cells, plasmablasts, myeloid cells, endothelial cells, PVL, and CAFs. The dot plot on the right shows *NR2F1*, *TGFB1*, *SOX9*, and *RARB* expression by cell type in the single-cell Cohort 2. Symbols in the figure are as follows: ns, *p* > 0.05; *, *p* ≤ 0.05; **, *p* ≤ 0.01; ****, *p* ≤ 0.0001.

**Figure 7 cancers-14-02962-f007:**
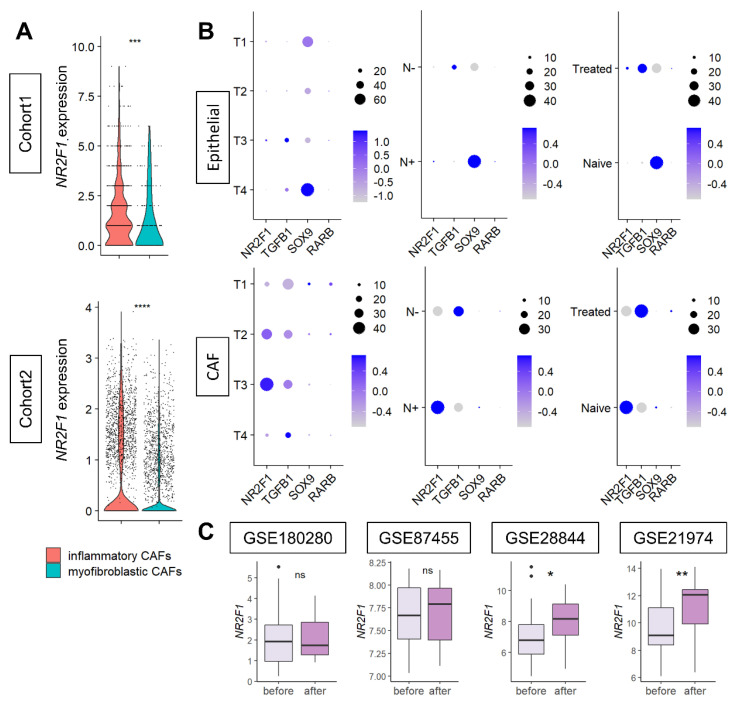
(**A**) Violin plots showing *NR2F1* expression of inflammatory CAF (iCAF) and myofibroblastic CAF (myCAF) of the two single−cell cohorts. One dot represents one cell. One−way ANOVA test was performed, and *** indicates *p* ≤ 0.001. (**B**) Dot plots showing *NR2F1*, *TGFB1*, *SOX9*, and *RARB* expressions in epithelial (cancer) cells and CAFs of single−cell Cohort 2 by T category of AJCC TNM staging, with and without lymph node metastasis, and with and without prior drug therapy. The size of each dot indicates the number of cells, and the purple intensity indicates the expression level. (**C**) Boxplots showing *NR2F1* expression before and after neoadjuvant chemotherapy in the four cohorts. Mann–Whitney U test was used for analysis, and the symbols in the figure are as follows: ns, *p* > 0.05; *, *p* ≤ 0.05; **, *p* ≤ 0.01, ****, *p* ≤ 0.0001.

## Data Availability

Publicly available datasets were analyzed in this study. TCGA data can be found here: (https://www.cbioportal.org, accessed on accessed on 21 October 2021. /Breast Invasive Carcinoma (TCGA, PanCancer Atlas)). METABRIC data can be found here: (https://www.cbioportal.org, accessed on 21 October 2021. /Breast Cancer (METABRIC, Curtis et al. [27]). GEO96058, GSE12276, GSE124647, GSE159956, GSE110590, GSE180280, GSE87455, GSE28844, GSE21974, and GSE172882 data can be downloaded from the Gene Expression Omnibus website with access number: (https://www.ncbi.nlm.nih.gov/geo, accessed on 2 November 2021). Two cohorts of single-cell seq data were downloaded from the Single-cell Portal (https://singlecell.broadinstitute.org/single_cell, accessed on 29 November 2021. (Wu et al. [46] and Wu et al. [47]).

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
