# Peer review of "NR2F1, a Tumor Dormancy Marker, Is Expressed Predominantly in Cancer-Associated Fibroblasts and Is Associated with Suppressed Breast Cancer Cell Proliferation"

_cancers, 2022, doi:10.3390/cancers14122962_

Round 1

Reviewer 1 Report

This manuscript adds evidence that NR2F1 is a dormancy marker. In addition, it shows that NR2F1 is primarily expressed in CAFs, particularly in inflammatory CAFs. Given that dormant disseminated tumor cells (DTCs) highly express NR2F1 (Fluegel et al. Nat Cell Biol 2017), it is puzzling how NR2F1 in CAFs of the primary tumor would contribute to dormancy of DTCs. It would be nice, if the authors would discuss this important point in more detail.

The authors choose chemotherapy as a treatment option to compare it with NR2F1 levels. It would be interesting to see, how NR2F1 levels would change with endocrine treatment and/or endocrine resistance.

Breast cancer is a heterogenous disease. Different subtypes behave differently in many aspects. It would be great to see some subtype-specific data.

The authors state “We demonstrated that the expression of NR2F1, RARB, and TGFB1 genes are higher in previously established dormant cells (D2OR murine breast cancer cells [44] compared to the proliferative cells (D2A1 cells) in  both 2D and 3D cultures (Figure 1A, all p < 0.02).” This is not true for NR2F1 in 2D. Please correct.

Reviewer 2 Report

The authors present interesting findings that a tumor dormancy marker, NR2F1, is predominantly expressed in the inflammatory CAFs, and high expression of NR2F1 is associated with suppressed immune response and increased density of stromal cells. However, this reviewer has a few concerns that need to be addressed before accepting this article for publication.

Major:

This reviewer noticed that all the analysis was performed on the public data sets of bulk RNA-seq or single cell sequencing of primary breast tumors. Can the authors do some analyses using data generated from both primary and metastatic tumors to check if there is any difference of NR2F1 expression, and how NR2F1 expression is correlated with metastasis?

As mentioned by the authors in Figure 3, there is no correlation between NR2F1 expression in the primary breast tumor and late recurrence. The authors need to analyze the single cell sequencing data and show whether the late recurrence in other organs, such as lung, bone, and brain, other than lymph node, is correlated with NR2F1 expression in the CAFs. Otherwise, the authors cannot make a conclusion that CAF-expressed NR2F1 regulates breast tumor dormancy.

Moreover, the authors need to provide some, at least minimum, evidence or clues that CAF-expressed NR2F1 is responsible for tumor dormancy regulation.

Minor:

Line 170, “Figure 1E” should be “Figure 1D”.

Round 2

Reviewer 1 Report

All my concerns have been satisfactorily addressed

Reviewer 2 Report

This reviewer's comments have been addressed.